Investigating the role of IDO1 in tumors: correlating IDO1 expression with clinical pathological features and prognosis in lung adenocarcinoma patients

Yin Zhidong 1 2
Sun Bohao 1
Wang Sisi 1
Xu Xi 1
Cheng Lu 1
Gao Yue gaoyue@hospital.westlake.edu.cn 3 4
Jin Er jiner@hospital.westlake.edu.cn 2
1 Department of Pathology, Second Affiliated Hospital, Zhejiang University School of Medicine , Hangzhou , Zhejiang Province , China
2 Department of Respiratory Medicine, Affiliated Hangzhou First People’s Hospital, Westlake University, School of Medicine , Hangzhou , Zhejiang Province , China
3 Department of Geriatric, Affiliated Hangzhou First People’s Hospital, Westlake University, School of Medicine , Hangzhou , Zhejiang Province , China
4 Zhejiang Provincial Key Laboratory of Traditional Chinese Medicine for the Prevention and Treatment of Major Chronic Disease in the Elderly , Hangzhou , Zhejiang Province , China
Uversky Vladimir
Electronic publication date: 2025 Feb 19
Publication date: 2025
Volume: 13
Electronic Location ID: e18776
Received 2024 Aug 19; Accepted 2024 Dec 9
Copyright: ©2025 Yin et al.
Copyright year: 2025
Copyright holder: Yin et al.
License: This is an open access article distributed under the terms of the Creative Commons Attribution License, which permits unrestricted use, distribution, reproduction and adaptation in any medium and for any purpose provided that it is properly attributed. For attribution, the original author(s), title, publication source (PeerJ) and either DOI or URL of the article must be cited.
License URL: https://creativecommons.org/licenses/by/4.0/

Keywords: IDO1, Bioinformatics, Lung adenocarcinoma, Expression, Function, Prognosis

Funding: Zhejiang Provincial Basic Public Welfare Research Program No. LY19H160031 Zhejiang Provincial Medicine and Health Science and Technology Development Plan Project No. 2019KY126 This work was supported by the Zhejiang Provincial Basic Public Welfare Research Program (No. LY19H160031) and the Zhejiang Provincial Medicine and Health Science and Technology Development Plan Project (No. 2019KY126). The funders had no role in study design, data collection and analysis, decision to publish, or preparation of the manuscript.

==============================
Purpose

This study aimed to investigate the role and expression patterns of IDO1 in various tumors, focusing on its correlation with clinical pathological characteristics and prognosis in patients specifically diagnosed with lung adenocarcinoma.

Methods

Pan-cancer analysis assessed IDO1 function across different tumor types. Bioinformatics tools, immunohistochemistry techniques, and statistical analyses were employed to evaluate IDO1 expression levels and their association with clinical pathological features and prognosis in patients with lung adenocarcinoma.

Results

IDO1 was found to be significantly overexpressed in various types of tumors, with higher levels correlating with poorer progression-free survival (PFS) and overall survival (OS). In lung adenocarcinoma patients, IDO1 protein was predominantly localized to the cytoplasm and cell membrane of tumor cells, with higher expression observed in tumor cells closer to normal lung tissue. Statistical analysis revealed no significant differences in IDO1 expression based on the patient’s clinical data, including gender, age, tumor location, allergy history, hypertension history, cardiovascular disease history, tumor history, diabetes (both type 1 and type 2), body mass index, smoking history, family history, alcohol history, and tumor maximum diameter (P > 0.05). However, IDO1 expression positively correlated with lymph node metastasis, pleural invasion, tumor recurrence, lower tumor differentiation, solid tumor components, preoperative chemotherapy, and clinical tumor, node, metastasis (TNM) staging (*P < 0.05), while negatively correlating with prior surgical history (*P < 0.05). Patients exhibiting high IDO1 expression levels demonstrated significantly worse PFS and OS (***P < 0.001 and **P = 0.003, respectively).

Conclusion

High IDO1 expression in lung adenocarcinoma correlates with increased tumor invasiveness, metastatic potential, advanced clinical stage, and poorer prognosis.

Introduction

Lung cancer stood as one of the most prevalent malignant tumors globally, with an incidence rate of 12.4% and a mortality rate of 18.7%, steadily increased year by year (Sung et al., 2021; Bray et al., 2024). Non-small cell lung cancer (NSCLC), comprising 80–85% of all cases, included adenocarcinoma (40%) and squamous cell carcinoma (30%) (Wu et al., 2024; Wei et al., 2023; Sahu et al., 2023). Early-stage symptoms often lacked specificity, resulting in advanced-stage diagnoses where surgical intervention was no longer viable. Consequently, the 5-year survival rate remained low, and systemic chemotherapy and radiotherapy offered limited efficacy (Liu et al., 2022; Cai, Dong & Wang, 2018; Ge et al., 2022). The advent of targeted therapy and immunotherapy introduced novel frontline treatment options for advanced-stage lung cancer patients. Examples included tyrosine kinase inhibitors (TKIs) that targeted the epidermal growth factor receptor (EGFR), and immune checkpoint inhibitors (ICIs) designed to block programmed cell death ligand 1 (PD-L1), both of which had demonstrated promise. However, EGFR-TKIs typically developed resistance in patients after one year, and ICIs exhibited greater efficacy in patients with high PD-L1 expression, with poorer responses seen in those with low or absent expression (Hanna et al., 2017; Nan et al., 2017; Pardoll, 2012; Reck et al., 2016). Therefore, exploring novel immune prognostic markers, identifying immunotherapy targets, and advancing research into new therapeutic agents become paramount research priorities in the field.

Indoleamine 2,3-dioxygenase 1 (IDO1) was a monomeric oxidoreductase and a rate-limiting enzyme in human tryptophan metabolism, responsible for more than 95% of tryptophan (Trp) degradation via the kynurenine (Kyn) pathway (Dolsak, Gobec & Sova, 2021; Zhang et al., 2022). The literature reported heightened IDO1 levels in numerous human tumors, often correlating with increased tumor aggressiveness and poorer patient outcomes (Platten et al., 2019; Yao et al., 2021; Mandarano et al. 2020; Bishnupuri et al., 2019). IDO1 inhibitors were demonstrated to exhibit significant immunotherapeutic efficacy in the treatment of malignant tumors such as metastatic melanoma and renal cell carcinoma (Zhang et al., 2021). Furthermore, studies indicated that IDO1 inhibitors could synergize with conventional cancer therapies and immune checkpoint inhibitors (Li et al., 2018). Moreover, 35 clinical trials were underway investigating various types of IDO1 inhibitors (Du et al., 2019).

In the literature related to lung cancer, we observed that, compared to adjacent normal lung tissues, the expression level of IDO1 was significantly upregulated in stage I, II, and III lung adenocarcinomas (Zhao et al., 2022). In patients with stage III NSCLC who were inoperable and had undergone chemotherapy and radiotherapy, high IDO1 expression was associated with poorer overall survival (OS) and progression-free survival (PFS) (Wu et al., 2022). In studies on IDO1 inhibitors, it was found that these inhibitors could enhance the functionality of NK cells while also inhibiting tumor growth in NSCLC mouse models (Fang et al., 2022). Additionally, IDO1 inhibitors were shown to potentiate the inhibitory effects of conventional fractionated radiotherapy on NSCLC tumors (Lan et al., 2023). When combined with nicotinamide phosphoribosyltransferase (NAMPT), IDO1 inhibitors were effective in treating chemotherapy- and targeted therapy-resistant NSCLC (Wang et al., 2023). These findings suggested that IDO1 might have potential as an immune prognostic biomarker and a novel therapeutic target for immunotherapy or combination therapy in lung cancer. Based on this, we planned to initially investigate the expression patterns and potential roles of IDO1 through bioinformatics analysis, followed by experimental validation to systematically assess its functional and clinical significance in NSCLC.

In this study, we utilized cancer genomics datasets from The Cancer Genome Atlas (TCGA), the Genotype-Tissue Expression (GTEx) project, and the Gene Expression Omnibus (GEO) databases to examine IDO1 expression across various human cancers and its association with cancer prognosis. Furthermore, we investigated the interactions between IDO1 protein and other human proteins. In preliminary experiments, we observed higher IDO1 expression in adenocarcinoma compared to other types of NSCLC. Consequently, we focused our analysis on IDO1 expression levels specifically in adenocarcinoma and its correlations with immune cell infiltration, gene mutations, immune checkpoint markers, and patient prognosis. To explore these relationships further, we conducted immunohistochemistry to evaluate IDO1 expression in lung adenocarcinoma tissues, assessing its potential implications for clinical-pathological characteristics and patient outcomes. Our objective was to evaluate IDO1’s viability as both an immune-related prognostic marker and a potential target for novel immunotherapeutic strategies in lung adenocarcinoma.

Materials and Methods

Pan-cancer analysis

Data from the TCGA database were downloaded and processed using specific filtering methods. The expression of IDO1 protein across multiple human cancers was analyzed using GEPIA (http://gepia.Cancer-pku.cn/index.html), TIMER 2.0 (https://cistrome.shinyapps.io/timer/), SangerBox (http://vip.sangerbox.com/home.html), Kaplan–Meier plotter (http://kmplot.com/analysis/index.php?p=background), and STRING (https://cn.string-db.org/cgi/input?sessionId=bAHDzo1bCrD3&input_page_show_search=on). This analysis investigated the association between high IDO1 protein expression in cancers and patient prognosis, as well as the functional interactions of human IDO1 protein with other proteins.

Bioinformatics analysis of IDO1 in lung adenocarcinoma

Utilizing data from TCGA, GTEx, and GEO databases, we analyzed IDO1 expression specifically in lung adenocarcinoma. The correlation between high IDO1 expression and patients’ progression-free survival and overall survival was investigated. Additionally, we analyzed the relationship between IDO1 expression levels and immune cell infiltration, gene mutations, and immune checkpoint markers in lung adenocarcinoma patients.

Data collection

Clinical information was collected from 144 patients diagnosed with lung adenocarcinoma between August 2015 and July 2017 at the Second Affiliated Hospital of Zhejiang University School of Medicine. Inclusion criteria required patients to have undergone pulmonary segmentectomy or lobectomy, with postoperative pathology confirming adenocarcinoma diagnosis, and to have relatively complete clinical-pathological and follow-up data. Ethical approval for this study ‘Potential of Targeting IDO1 as a Treatment for Non-Small Cell Lung Cancer’ was obtained from the Clinical Research Ethics Committee of The Second Affiliated Hospital, Zhejiang University School of Medicine (approval No. 2024-0552; Hangzhou, China). The Clinical Research Ethics Committee also waived the requirement for informed consent. In this study, the follow-up period for the majority of patients occurred in 2022.

Immunohistochemistry staining

Immunohistochemical techniques were employed to conduct stain analysis on tumor tissues obtained from 144 patients diagnosed with lung cancer. The IDO1 antibody (catalog number NBP1-87702), sourced from Novus Biologicals, LLC, was diluted at a ratio of 1:1,000 for this analysis. Secondary antibodies, DAB chromogen solution, hematoxylin staining solution, PBS buffer, and citric acid antigen repair solution were obtained from Beijing Zhongshan Golden Bridge Biotechnology Co., Ltd.

Results interpretation

Staining results were collectively interpreted by three pathologists, who assigned scores based on the following criteria: tumor intensity was graded as 0 (negative), 1 (weak positive), 2 (moderate positive), or 3 (strong positive); tumor extent was graded as 0 (0–10%), 1 (10%–25%), 2 (26%–50%), 3 (51%–75%), or 4 (76%–100%). The final score was calculated by multiplying the intensity score by the extent score. Scores ranging from 6 to 12 were considered indicative of high IDO1 expression or positive, whereas scores ranging from 0 to 5 points indicated low expression or absence of IDO1, hence negative.

Statistical analysis

Data from scoring results and clinical cases were compiled in Excel and analyzed using SPSS 26.0 and GraphPad Prism 10 software. Chi-square tests were employed to evaluate the relationship between IDO1 expression and clinical data in NSCLC patients. Correlation analyses were conducted to explore associations between IDO1 expression and clinical parameters. Survival curve analysis was utilized to illustrate the relationship between IDO1 expression levels and both progression-free survival and overall survival among patients. Statistical significance was defined as *P < 0.05.

Results

Pan-cancer analysis results

IDO1 exhibited high expression across various types of cancer patients (Figs. 1A and 1D). Cancers characterized by high IDO1 expression exhibit decreased progression-free survival (PFS) and overall survival (OS) rates (Fig. 2), highlighting a negative prognostic impact associated with elevated IDO1 expression in cancer patients. Furthermore, there existed a correlation between the functional activity and expression of human IDO1 protein with several other human proteins, including TDO2, KYNU, and others (Figs. 1B and 1C).

Figure 1 Pan-cancer IDO1 expression and role.

(A) IDO1 expression in various cancers and paired normal tissues from the SangerBox database (*P < 0.05, **P < 0.01, ***P < 0.001). (B) Network diagram illustrating the relationship between IDO1 and other human proteins from the STRING database. (C) Correlation scores between IDO1 and other human proteins from the STRING database. (D) IDO1 expression in various cancers and paired normal tissues from the GEPIA database. (GBM, Glioblastoma multiforme; GBMLGG, Glioma; LGG, Brain Lower Grade Glioma; UCEC, Uterine Corpus Endometrial Carcinoma; BRCA, Breast invasive carcinoma; CESC, Cervical squamous cell carcinoma and endocervical adenocarcinoma; ESCA, Esophageal carcinoma; STES, Stomach and Esophageal carcinoma; KIRP, Kidney renal papillary cell carcinoma; KIPAN, Pan-kidney cohort; COAD, Colon adenocarcinoma; COADREAD, Colon adenocarcinoma/Rectum adenocarcinoma Esophageal carcinoma; PRAD, Prostate adenocarcinoma; STAD, Stomach adenocarcinoma; HNSC, Head and Neck squamous cell carcinoma; KIRC, Kidney renal clear cell carcinoma; LIHC, Liver hepatocellular carcinoma; SKCM, Skin Cutaneous Melanoma; BLCA, Bladder Urothelial Carcinoma; OV, Ovarian serous cystadenocarcinoma; PAAD, Pancreatic adenocarcinoma; TGCT, Testicular Germ Cell Tumors; UCS, Uterine Carcinosarcoma; PCPG, Pheochromocytoma and Paraganglioma; ACC, Adrenocortical carcinoma; KICH, Kidney Chromophobe; DLBC, Lymphoid Neoplasm Diffuse Large B-cell Lymphoma; READ, Rectum adenocarcinoma; THYM, Thymoma).

Figure 2 Kaplan–Meier survival curve of human cancers with high and low IDO1 expression analyzed by the SangerBox database (A–H), the GEPAI database (I, J), and the Kaplan–Meier plotter database (K, L).

(A–H) High IDO1 expression was associated with worse DSS and OS in cohorts of GBMLGG, KIRP, LGG, and UVM. (I, J) High IDO1 expression was associated with worse DFS and OS in cohorts of KICH. (K, L) High IDO1 expression was associated with worse EFS and OS in cohorts of myeloma. (DSS, disease-specific survival; DFS, disease-free survival; EFS, event-free survival; OS, overall survival; GBMLGG, Glioma; KIRP, Kidney renal papillary cell carcinoma; LGG, Brain Lower Grade Glioma; KICH, Kidney Chromophobe; UVM, Uveal Melanoma).

Bioinformatics analysis results

IDO1 expression was higher in lung adenocarcinoma compared to normal lung tissue (Fig. 3A). Patients with high IDO1 expression in adenocarcinoma showed reduced first progression survival (FP) (Fig. 3B) and OS (Figs. 3C and 3D). Moreover, IDO1 expression in adenocarcinoma correlated with various immune cell infiltrations (Fig. 3E), multiple gene mutations including EGFR (Fig. 3F), and the expression of multiple immune checkpoint genes such as CD274 (Fig. 3G).

Figure 3 The expression and function of IDO1 in lung adenocarcinoma, as well as its relationship with prognosis.

(A) IDO1 expression in lung adenocarcinoma and paired normal tissue from the GEPIA database. (B) High IDO1 expression in lung adenocarcinoma was associated with worse first progression survival in the Kaplan–Meier plotter database. (C, D) High IDO1 expression in lung adenocarcinoma was associated with worse OS in the GSE31210 and GSE50081 datasets. (E) Correlations between IDO1 expression in lung adenocarcinoma and immune cells in the TIMER 2.0 database. (F) Correlations between IDO1 expression in lung adenocarcinoma and gene mutations in the SangerBox database. (G) Correlations between IDO1 expression in lung adenocarcinoma and immune checkpoints in the SangerBox database (LUAD, Lung adenocarcinoma).

Expression and localization of IDO1 in lung adenocarcinoma

Microscopic examination revealed that tumor cells with high IDO1 protein expression were diffusely distributed within tumor tissues. IDO1 protein was prominently localized in the cytoplasm and membrane, exhibiting a brown-yellow granular staining pattern in the cytoplasm (Fig. 4). Additionally, heterogeneous expression of IDO1 was observed among tumor cells, with those closer to normal lung tissue exhibiting higher IDO1 expression levels (Figs. 5A and 5B).

Figure 4 Representative images of hematoxylin and eosin (HE) staining and IDO1 immunohistochemical positive staining of different subtypes of lung adenocarcinoma (100x magnification).

(A) HE staining and IDO1 immunohistochemical positive staining in acinar, papillary, and solid subtypes of lung adenocarcinoma. (B) HE staining and IDO1 immunohistochemical positive staining in micropapillary, cribriform, and solid subtypes of lung adenocarcinoma.

Figure 5 The expression of IDO1 in lung adenocarcinoma and its correlation with clinicopathological features and patient prognosis.

(A, B) Tumor cells closer to normal lung tissue exhibit higher expression of IDO1 in lung adenocarcinoma (40x magnification). (C–G) Higher IDO1 scores were observed in patients with lymph node metastasis, recurrence, poorly differentiated tumor cells, preoperative chemotherapy, or higher clinical TNM staging (*P < 0.05, **P < 0.01, ***P < 0.001). (H) High IDO1 expression was associated with worse progression-free survival (PFS) in patients with lung adenocarcinoma (***P < 0.001). (I) High IDO1 expression was associated with worse overall survival (OS) in patients with lung adenocarcinoma (**P = 0.003).

IDO1 positive staining and clinical-pathological data analysis in lung adenocarcinoma patients

Non-significant statistical clinical-pathological data

Among adenocarcinoma patients, IDO1 positive staining was observed in 41 cases, while 103 cases exhibited negative staining. Chi-square tests indicated no statistically significant differences between high IDO1 expression and patient characteristics, including gender, age, tumor location, history of allergies, hypertension, cardiovascular diseases, tumor history, diabetes (both type 1 and type 2), body mass index, smoking history, family history, alcohol consumption history, and maximum tumor diameter (P > 0.05). Refer to Table 1 for detailed information.

Table 1 IDO1 expression and clinical-pathological data analysis in lung adenocarcinoma patients.

Non-statistically significant clinical data	Pos (41)	Neg (103)	c2 value	p value	Statistically significant clinical data	Pos (41)	Neg (103)	c2 value	p value	r value	p value	
Sex	Male	20	44	0.436	0.509	Lymph node metastasis	Yes	25	35	8.793	0.003	0.247	0.003	
Female	21	59	No	16	68	
Age	≥65	19	37	1.340	0.247	Pleural invasion	Yes	24	37	6.142	0.013	0.207	0.013	
<65	22	66	No	17	66	
Tumor localization	Lift	16	49	0.865	0.352	Surgical history	Yes	4	28	5.154	0.023	−0.189	0.023	
Right	25	54	No	37	75	
Allergic history	Yes	8	24	0.244	0.622	Recurrence and distant metastasis	Yes	17	25	4.195	0.041	0.171	0.041	
No	33	79	No	24	78	
Hypertension	Yes	17	39	0.160	0.689	Tumor differentiation grade	Low	17	19	8.286	0.004	0.240	0.004	
No	24	64	Middle or High	24	84	
Cardiovascular disease	Yes	7	28	1.630	0.202	Tumour with solid components	Yes	14	15	6.993	0.008	0.220	0.008	
No	34	75	No	27	88	
Tumor history	Yes	2	13	1.884	0.170	Preoperative chemotherapy	Yes	6	5	3.976	0.046	0.166	0.047	
No	39	90	No	35	98	
Diabetes	Yes	4	7	0.364	0.546	Clinical TNM staging	I	13	55	10.571	0.014	0.241	0.004	
No	37	96	II	10	24	
BMI	≥25	11	27	0.006	0.940	III	16	24	
<25	30	76	IV	2	0	
Smoking history	Yes	17	31	1.705	0.192	PFS	≥36 Months	15	64	8.474	0.004	−0.244	0.004	
No	24	72	<36 Months	26	37	
Family history	Yes	5	10	0.194	0.659	OS	≥60 Months	17	65	6.264	0.012	−0.210	0.012	
No	36	93	<60 Months	24	36	
Drinking history	Yes	10	32	0.633	0.426		
No	31	71								
Tumor maximum diameter	≥30 mM	14	39	0.174	0.676		
<30 mM	27	64								

Statistically significant clinical-pathological data

In adenocarcinoma patients, 41 cases showed IDO1 positive staining while 103 cases exhibited negative staining. Chi-square tests revealed significant differences in high IDO1 expression concerning lymph node metastasis, pleural invasion, surgical history, recurrence, tumor cell differentiation, clinical tumor, node, metastasis (TNM) staging, preoperative chemotherapy status, progression-free survival, and overall survival (*P < 0.05). Correlation analysis indicated that high IDO1 expression positively correlated with lymph node metastasis, pleural invasion, recurrence, poorly differentiated tumors, solid components, preoperative chemotherapy, and TNM staging, while it negatively correlated with patient surgical history, PFS, and OS (*P < 0.05). Refer to Table 1 for detailed information. A comparison of clinical data with IDO1 immunohistochemical scores is illustrated in Figs. 5C–5G below.

High IDO1 expression and prognostic analysis in lung adenocarcinoma patients

In survival curve analysis, we observed that patients with high IDO1 expression exhibited significantly lower progression-free survival (***P < 0.001) and overall survival (**P = 0.003) rates (Figs. 5H and 5I).

Discussion

IDO1 was a monomeric oxidoreductase that contained heme and comprised two structural domains. The large C-terminal domain (CTD) was essential for IDO1’s catalytic activity, whereas the small N-terminal domain (NTD) primarily served signaling functions. IDO1 was widely distributed across various human tissues and exhibited high expression in antigen-presenting cells such as dendritic cells, monocytes, and macrophages (Zhai et al., 2018; Merlo, Peng & Mandik-Nayak, 2022). Functioning as a rate-limiting enzyme, IDO1 played pivotal biological roles by modulating the levels of tryptophan and kynurenine metabolites. IDO1 was involved in the suppression of CD8+ T effector cells and natural killer (NK) cells, while enhancing the activity of CD4+ regulatory T cells (Tregs), dendritic cells (DCs), and myeloid-derived suppressor cells (MDSCs) (Du et al., 2019). IDO1 was involved in peripheral immune tolerance, contributing to maintaining homeostasis by preventing autoimmunity or immunopathology that would result from uncontrolled and overreacting immune responses (Pallotta et al., 2022; Tang et al., 2022). However, overexpression of IDO1 has also been implicated broadly in human pathology, including exacerbating pathogen infections, contributing to poor prognosis in various cancers, promoting autoimmunity, increasing neurodegenerative conditions, and more (Pallotta et al., 2022; Li et al., 2017; Taleb, 2019).

The literature reported that in various types of cancers, including ovarian cancer, colorectal cancer, endometrial cancer, melanoma, cervical cancer, glioblastoma, lung adenocarcinoma, and diffuse large B-cell lymphoma, overexpression of IDO1 in tumor cells or other cells within the tumor microenvironment (TME) was associated with more aggressive cancer phenotypes, advanced disease stages, and poorer clinical outcomes (Platten et al., 2019; Yao et al., 2021; Li et al., 2017; Liu et al., 2018). In our pan-cancer analysis, we also found that IDO1 was highly expressed in multiple human cancers, and cancers with high IDO1 expression—such as GBMLGG, KIRP, LGG, KICH, UVM, and myeloma, among others—demonstrated significantly lower PFS and OS. These findings suggested that high IDO1 expression correlates with increased tumor invasiveness and poor patient prognosis, indicating the potential of IDO1 as a prognostic biomarker and a novel therapeutic target for immune treatment in various cancers. Our research team primarily focused on targeted and immunotherapy approaches for lung cancer. Moreover, the preliminary experiments revealed that IDO1 expression was higher in lung adenocarcinoma compared to other types of NSCLC. Therefore, we focused on analyzing the role of IDO1 in lung adenocarcinoma to explore its potential as an immune prognostic biomarker and a new target for immunotherapy.

In the bioinformatics analysis, we found that IDO1 expression in lung adenocarcinoma was higher than in normal lung tissue. Additionally, patients with high IDO1 expression in adenocarcinoma exhibited lower PFS and OS, which was consistent with reports in the literature (Zhao et al., 2022; Jin et al., 2024). This finding was further validated in subsequent experiments. According to literature (Zhao et al., 2022; Bessede et al., 2023; Schalper et al., 2017; Wu et al., 2023; Kozuma et al., 2018), in NSCLC, including lung adenocarcinoma, clinical TNM staging showed a negative correlation with IDO1 expression. In later stages of clinical progression, IDO1 expression in tumors was found to be elevated, but there was a marked reduction in the infiltration of T lymphocytes, especially CD8+ T cells. The bioinformatics analysis further revealed that IDO1 expression in adenocarcinoma was negatively correlated with tumor heterogeneity, while positively correlated with various immune cell types, particularly with immune suppressive cells such as DCs. These findings suggested that in lung adenocarcinoma, IDO1 might promote tumor immune evasion by inhibiting the infiltration and survival of cytotoxic T cells within the tumor microenvironment, while enhancing the activity of immunosuppressive cells like Tregs and DCs.

According to literature (Zhao et al., 2022; Wu et al., 2022), IDO1 expression was higher in stage III lung cancer compared to stages I and II, and the activity of IDO1 in stage III lung cancer was negatively correlated with OS and PFS. Mandarano et al. (2021) reported that the Kyn/Trp ratio could also serve as a clinical pathological marker for prognosis in NSCLC patients, monitoring cancer invasiveness and progression. Additionally, Pan et al. (2017) indicated that IDO1 expression in lung cancer patients was associated with microvascular density (MVD) marked by CD34 and CD146 and that silencing of IDO1 could inhibit tumor angiogenesis. In the present study, we found that IDO1 expression was positively correlated with lymph node metastasis, pleural invasion, recurrence and metastasis, poor tumor differentiation, solid components, and TNM staging in adenocarcinoma patients. It was also negatively correlated with PFS and OS. Microscopic examination revealed that tumors expressing IDO1 exhibited heterogeneous expression, with tumor cells near normal lung tissue showing higher levels of expression. These findings suggested that IDO1 played a crucial role in immune evasion, neoangiogenesis, and metastasis in lung adenocarcinoma. Furthermore, tumors with positive IDO1 expression demonstrate increased invasiveness and metastatic potential, leading to a higher clinical stage and poorer prognosis for patients. Therefore, we believe that IDO1 holds potential as an immunological prognostic marker for lung adenocarcinoma.

Bioinformatics analysis revealed that the expression of IDO1 was negatively correlated with mutations in genes such as EGFR, while positively correlated with the immune checkpoint protein CD274 (also known as PD-L1). In a previously published study (Jin et al., 2024), we demonstrated that IDO1 inhibitors could suppress the expression of PD-L1 and p-EGFR in the H1975 cell line. In related literature, a co-expression of IDO1 and PD-L1 had been identified, with co-expressing lung adenocarcinomas exhibiting more aggressive characteristics (Kozuma et al., 2018). Dual inhibition of MEK and PD-L1 had been shown to enhance IDO1 expression in tumors, leading to immune escape, however, this effect could be reversed by the addition of IDO1 inhibitors (Della Corte et al., 2022). Furthermore, combination therapy with IDO1 inhibitors and nicotinamide phosphoribosyltransferase (NAMPT) was proposed for the treatment of chemotherapy- and targeted therapy-resistant NSCLC (Wang et al., 2023). These findings suggested that IDO1 interacted with various gene mutations and immune checkpoints, with particular emphasis on its potential synergistic effect with PD-L1. Based on these findings, IDO1 held promise as a new therapeutic target for combination immunotherapy or targeted therapy in lung adenocarcinoma.

This study, in contrast to previous research, incorporated databases such as TCGA for exploratory analysis. Through bioinformatics analysis and experimental validation, we systematically evaluated the role of IDO1 in lung adenocarcinoma. Compared to earlier studies, our study included a larger and more detailed collection of clinical data and provided a more comprehensive description of IDO1 expression and localization under the microscope. For the first time, we reported that IDO1 expression was heterogeneous within tumors, with higher expression in tumor cells adjacent to normal lung tissue compared to other tumor cells. Additionally, IDO1 expression was associated with lower tumor differentiation and the presence of solid components. In this study, we also found that lung adenocarcinoma patients who underwent neoadjuvant chemotherapy exhibited higher IDO1 expression, a finding not previously reported in the literature. This might suggest that chemotherapy drugs could stimulate IDO1 expression in tumor cells, warranting further investigation.

Conclusions

Our study revealed uniform cytoplasmic and membranous expression of IDO1 protein in tumor cells and higher IDO1 expression observed in tumor cells near normal lung tissue. Moreover, high IDO1 expression in adenocarcinoma patients positively correlated with lymph node metastasis, pleural invasion, tumor recurrence, lower tumor differentiation grade, presence of solid components, and advanced clinical TNM staging. Conversely, there was a negative association with patient surgical history, PFS, and OS. These findings suggest that tumors exhibiting elevated IDO1 levels demonstrate increased invasiveness and metastatic potential, often presenting at more advanced clinical stages and associated with poorer prognosis. Additionally, our observations indicated higher IDO1 expression in lung adenocarcinoma tumors following chemotherapy. This study underscores that IDO1 was a promising immune prognostic marker and a potential target for novel immunotherapeutic approaches in lung adenocarcinoma.

Supplemental Information

Supplemental Information 1 Raw Data

We appreciate ChatGPT’s assistance with the English editing of this article.

Additional Information and Declarations

Competing Interests

Author Contributions

Human Ethics

Data Availability

The authors declare there are no competing interests.

Zhidong Yin conceived and designed the experiments, performed the experiments, analyzed the data, prepared figures and/or tables, authored or reviewed drafts of the article, and approved the final draft.

Bohao Sun conceived and designed the experiments, analyzed the data, prepared figures and/or tables, authored or reviewed drafts of the article, and approved the final draft.

Sisi Wang performed the experiments, prepared figures and/or tables, and approved the final draft.

Xi Xu analyzed the data, authored or reviewed drafts of the article, and approved the final draft.

Lu Cheng performed the experiments, prepared figures and/or tables, and approved the final draft.

Yue Gao conceived and designed the experiments, authored or reviewed drafts of the article, and approved the final draft.

Er Jin conceived and designed the experiments, authored or reviewed drafts of the article, and approved the final draft.

The following information was supplied relating to ethical approvals (i.e., approving body and any reference numbers):

Clinical Research Ethics Committee of The Second Affiliated Hospital, Zhejiang University School of Medicine

The following information was supplied regarding data availability:

The bioinformatics data is available at the GEPIA, TIMER 2.0, SangerBox, Kaplan-Meier plotter, and STRING databases and the raw data is available in the Supplemental File.

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
