# Peer review of "Investigating the role of IDO1 in tumors: correlating IDO1 expression with clinical pathological features and prognosis in lung adenocarcinoma patients"

_PeerJ, doi:10.7717/peerj.18776_

## Round 0.1 · original submission · Major Revisions

Please address issues pointed by both reviewers and amend manuscript accordingly.

Reviewer 1 ·

Basic reporting

Zhidong Yin et al. conducted a preliminary study on the role of IDO1 in patients with lung adenocarcinoma. The study made use of TCGA database and self-built specimen bank to make a beneficial exploration of this theme.

In the introduction, it is expected that the authors can provide more research background, IDO1 data related to the prognosis of lung adenocarcinoma patients have been reported in the past

Experimental design

The author uses several databases for comparison and verification, and the results are reliable. But novelty needs further emphasis.

Validity of the findings

1.IDO1 data related to prognosis in patients with lung adenocarcinoma have been previously reported (eg:IDO1 Expression and Prognosis in Different Clinical Stages of Lung Adenocarcinomaļ¼›Co-Expression of IDO1 and PD-L1 Indicates More Aggressive Features of Lung Adenocarcinoma)
If you can, please highlight the differences, advantages and novelty of this study and previous studies.

Additional comments

Does Figure 1 involve copyright issues?

The three pictures in Figure 2 seem to be repeating the same phenomenon. Please consider streamlining, pancancer abbreviations should be annotated in the figure notes.

Figure 3 also has the same problem. Multiple pictures seem to repeat similar phenomena, so a moderate simplification can be considered.

·

Basic reporting

none

Experimental design

none

Validity of the findings

none

Additional comments

This study employed a comprehensive research approach combining pan-cancer analysis and specific lung adenocarcinoma investigation to explore IDO1's expression patterns, clinical correlations, and prognostic significance. The innovation lies in its systematic evaluation of IDO1's role in lung adenocarcinoma through both bioinformatics analysis and experimental validation. While the study design is reasonable and the results are credible, several aspects require improvement before publication.
1. The introduction section needs a more comprehensive literature review of IDO1's current research status in lung cancer. The authors should supplement the latest progress of relevant studies, especially regarding IDO1's role in immunotherapy resistance and its potential as a therapeutic target, to better establish the research significance.
2. The results demonstrate that high IDO1 expression correlates with poor survival outcomes and various clinical-pathological features. However, the underlying mechanisms require deeper exploration. The discussion section should elaborate on the pathophysiological mechanisms by which IDO1 influences patient prognosis, particularly its interactions with immune cell infiltration, tumor microenvironment, and other immune checkpoints.
3. The article reveals the abnormal expression and prognostic significance of IDO1 in various solid tumors through pan-cancer analysis. However, the authors do not adequately explain the necessity and importance of pan-cancer analysis for the subsequent study of IDO1's role in lung adenocarcinoma. There is a lack of effective connection between the results of the pan-cancer analysis and the focused analysis of lung adenocarcinoma, making the logic between the two parts less coherent.

---

## Round 0.2 · accepted · Accept

All of the reviewers' comments were adequately addressed and revised manuscript is acceptable now.

Reviewer 1 ·

Basic reporting

no comment

Experimental design

no comment

Validity of the findings

no comment

Additional comments

no comment